# The force required to remove tubulin from the microtubule lattice by pulling on its α-tubulin C-terminal tail

Yin-Wei Kuo[1], Mohammed Mahamdeh [2,3], Yazgan Tuna[1] & Jonathon Howard [1✉]

Severing enzymes and molecular motors extract tubulin from the walls of microtubules by exerting mechanical force on subunits buried in the lattice. However, how much force is needed to remove tubulin from microtubules is not known, nor is the pathway by which subunits are removed. Using a site-specific functionalization method, we applied forces to the C-terminus of α-tubulin with an optical tweezer and found that a force of ~30 pN is required to extract tubulin from the microtubule wall. Additionally, we discovered that partial unfolding is an intermediate step in tubulin removal. The unfolding and extraction forces are similar to those generated by AAA-unfoldases. Lastly, we show that three kinesin-1 motor proteins can also extract tubulin from the microtubule lattice. Our results provide the first experimental investigation of how tubulin responds to mechanical forces exerted on its α-tubulin C-terminal tail and have implications for the mechanisms of severing enzymes and microtubule stability.

[1] Department of Molecular Biophysics and Biochemistry, Yale University, New Haven, CT, USA. [2] Harvard Medical School, Boston, MA, USA. [3] Cardiovascular Research Center, Massachusetts General Hospital, Boston, MA, USA. ✉email: joe.howard@yale.edu

The microtubule cytoskeleton serves as a network of tracks for motor proteins and provides mechanical support to eukaryotic cells[1]. The dynamical properties of the cytoskeleton allow cells to move, divide and change shape[2–4]. The microtubule cytoskeleton is subject to external mechanical forces —from the environment or from active processes within or outside the cell[5–7]—and to internal forces, which are generated by motors[8], by microtubule-associated proteins[9,10], or by severing proteins such as spastin and katanin[11]. The integrity of the microtubule cytoskeleton depends on the stability of microtubules in the presence of these mechanical forces. While we have some knowledge of how much force is needed to remove tubulin dimers from the ends of a microtubule[12–15], it is unknown how much force is required to remove tubulin dimers from the wall of the microtubule.

Microtubule-severing proteins are thought to pull tubulin subunits out of the microtubule lattice by exerting mechanical forces on the C-terminal tails (CTTs) of the tubulin;[16] the microtubule filament is thought to break when enough tubulin subunits have been removed from the lattice. The current model of severing, based on structural similarity to other AAA + unfoldases and disaggregases such as Hsp100[17,18], ClpX[19,20] and Vps4[21,22] that unfold or disassemble proteins (or protein complexes), is that severases use energy derived from ATP-hydrolysis to produce mechanical forces that pull proteins through a central channel made by the six AAA domains[11,23]. Biochemical and structural studies show that severing enzymes bind tubulin CTTs[24–26], but experimental observations of the force generation and tubulin extraction steps are absent.

In addition to severing enzymes, molecular motors such as kinesins and dynein can also remove tubulin subunits from the shafts of microtubules as they walk[27–29]. This removal of tubulin dimers from the internal lattice is distinct from the well-known activities of motors removing tubulin from microtubule ends[30]. Bending of microtubules by fluid flow can also cause dissociation of tubulin subunits from the lattice[31]. Thus, several lines of evidence suggest that mechanical forces can facilitate the removal of tubulin subunits from the microtubule shaft[32]. However, direct experimental investigation of the tubulin extraction process is lacking, due, in part, to the difficulty of applying force to a single tubulin subunit in a site-specific manner.

Here, we address two central questions concerning the tubulin extraction process. First, how much force is needed to pull out a tubulin subunit? And second, is tubulin unfolded during the pulling process?

## Results

**Development of a site-specific labeling method.** To apply forces to tubulin, we developed a strategy to introduce a functional handle specific to the C-terminus of α-tubulin by exploiting the broad substrate tolerance of tubulin-tyrosine ligase (TTL) reported previously[33,34]. The tubulin CTTs are exposed on the outer surface of the microtubule and are the proposed pulling sites of severing enzymes. Thus, tubulin CTTs are suitable handles through which to apply mechanical forces. TTL catalyzes the covalent addition of tyrosine and tyrosine analogs to the C-terminus of detyrosinated α-tubulin. We expressed and purified human TTL with an N-terminal His₆-SUMO tag (Supplementary Figure 1). The tyrosination activity of purified TTL was verified by the increase of tyrosinated tubulin after incubating TTL with bovine brain tubulin in the presence of ATP and L-tyrosine (Fig. 1a). We introduced a bio-orthogonal azide group to the C-terminus of α-tubulin by using 3-azido-tyrosine as the substrate for TTL. The desired functional molecules containing cycloalkyne were covalently conjugated by strain-promoted

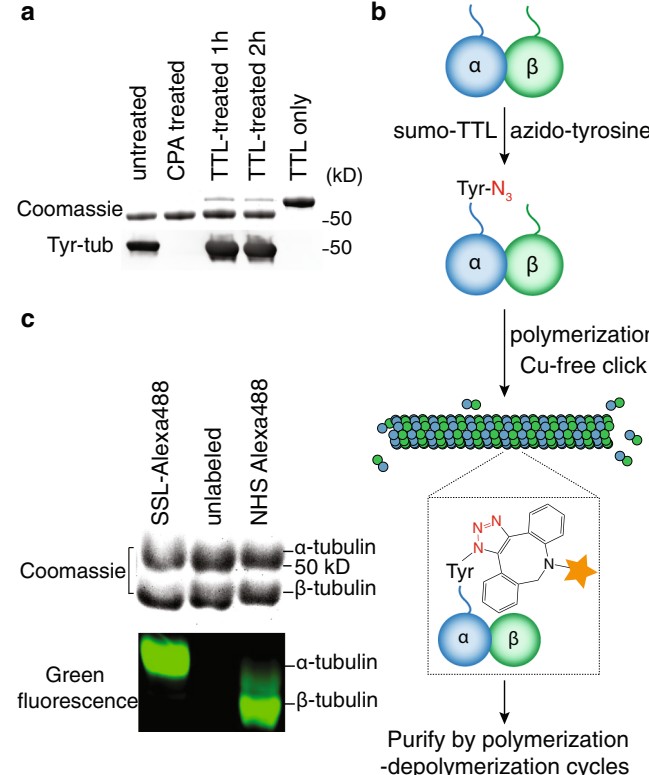

**Fig. 1 Site-specific functionalization of tubulin. a** Recombinant tubulin-tyrosine ligase (TTL) increased the tyrosination of bovine-brain tubulin. The tyrosinated-tubulin level detected by antityrosinated tubulin antibody (clone YL1/2) increased after 1 h of incubation (bottom panel). Carboxypeptidase A (CPA) cleaves the C-terminal tyrosine of α-tubulin; CPA-treated tubulin therefore serves as a negative control. Experiments were repeated with duplicates. **b** Schematic process of the site-specific labeling and purification of tubulin. The orange star represents the functional moieties of interest including fluorophores, affinity tags (e.g., biotin) and macromolecules (e.g., oligo-DNAs). **c** High-resolution SDS-PAGE of Alexa-Fluor-488-labeled tubulin with either site-specific labeling (SSL) or non-specific amine-reactive labeling (labeled with NHS ester of Alexa Fluor 488). SSL-tubulin showed fluorescence signal only on α-tubulin, while tubulin labeled with NHS ester showed conjugation on both tubulin chains with a higher fluorescence signal on β-tubulin, potentially due to the higher reactivity of the β-tubulin lysine side chains. Experiments were repeated with duplicates. Uncropped gels and Western blots were included in Supplementary Figure 8.

azide-alkyne cycloaddition (SPAAC) (Fig. 1b). To test the labeling specificity, we first conjugated Alexa Fluor 488 using this site-specific labeling (SSL) method, and separated α- and β-tubulin by high-resolution SDS-PAGE. We found that the fluorophore conjugated with the SSL method is specific to α-tubulin (Fig. 1c), as previously shown[33]. We also found that the site-specific fluorophore labeling had little effect on the microtubule dynamic properties, even when ~43% of α-tubulin was labeled (Supplementary Figure 2a, b). Using SSL-biotinylated tubulin, we further confirmed that the labeling site is indeed on the C-terminus of α-tubulin using tandem mass spectrometry (Supplementary Figure 2c). Thus, we have constructed a site-specific labeling method to covalently conjugate a functional handle to the α-tubulin CTT using recombinant TTL and commercially available reagents.

**Pulling tubulin subunits with an optical tweezer.** We next developed a tubulin-pulling assay using optical tweezers (Fig. 2). Single-stranded oligo-DNA (a GC-rich 40-mer) was covalently

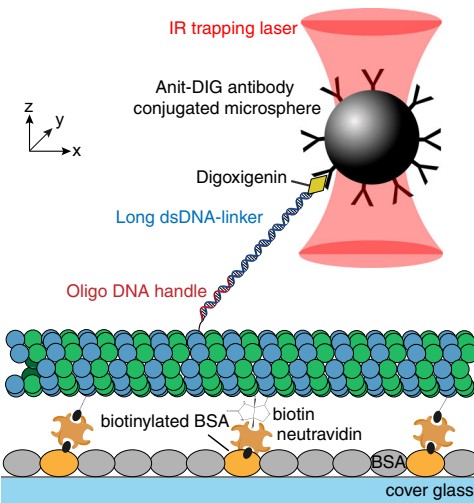

**Fig. 2 Experimental setup of the tubulin pulling assay using an optical trap.** Tubulin labeled with an oligo-DNA handle using the SSL method was linked to a long DNA linker through hybridization. Tubulin conjugated to the DNA-linker was polymerized with biotinylated tubulin and affixed to the surface of neutravidin-coated coverslips. The anti-digoxigenin antibody-coated microspheres were then bound to the DNA-linker labeled with digoxigenin. The mechanical force was exerted onto the α-tubulin CTT by pulling the microsphere with the optical trap along the y-axis, which is perpendicular to the long axis of microtubule (x-axis).

conjugated to the α-tubulin CTT with the aforementioned labeling method (Supplementary Figure 3a). A long, double-stranded DNA-linker (8.2 kb, 2.8-μm contour length) containing a 5'-flanking region that is complementary to the oligo-DNA handle was then hybridized to the oligo-DNA. Because the DNA linker and the microtubule surface are both negatively charged, it is unlikely that they strongly interact or that the DNA linker strongly influences the mechanical stability of the lattice. The other end of the double-stranded DNA linkers was labeled with digoxigenin, which was bound to a polystyrene microsphere coated with antidigoxigenin antibody. We attached the DNA linkers to taxol-stabilized microtubules, which was confirmed by total-internal-reflection-fluorescence (TIRF) microscopy and interference-reflection microscopy (IRM) (Supplementary Figure 3b). Taxol-stabilized microtubules were used because their high stability against depolymerization is compatible with the low-throughput tweezer experiments; the GDP microtubules used in the motor experiments described below are susceptible to breakage and spontaneous depolymerization, making them short-lived and unsuitable for the tweezer assay. To prevent the microtubule filaments from bending or sliding when pulled by the optical trap, the microtubules were copolymerized with biotinylated tubulin. These microtubules were affixed tightly to the coverslip of a flow-channel coated with neutravidin. Anti-digoxigenin antibody-coated microspheres were then introduced to form DNA-tethers. The binding of beads to the DNA linkers was confirmed by their tethered Brownian motions[35] around the microtubules.

We pulled single tubulin subunits in the lateral direction parallel to the coverslip surface and perpendicular to the microtubule axis using an optical trap (along the y-axis in Fig. 2). We chose this pulling direction mainly to avoid the attachment of more than one DNA linker to the bead during the pulling experiment. An example of a single-molecule force-extension trace is shown in Fig. 3a (median-filtered trace in orange). The low-force region of the force-extension curves corresponds to the stretching of the long double-stranded DNA linkers and was well described by the worm-like

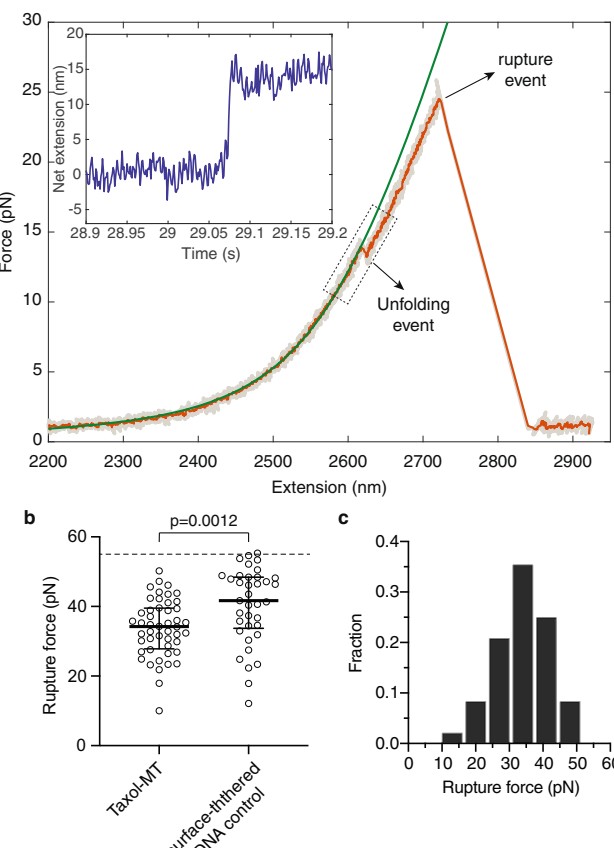

**Fig. 3 Pulling tubulin subunits from the taxol-stabilized microtubule lattice with an optical trap. a** Example of a force-extension curve from the tubulin-pulling assay. Unfolding and rupture events are indicated by arrows. Experiments were performed with a constant pulling velocity of 0.32 μm/s, using an 8.2 kb-DNA-linker (contour length 2.8 μm). Gray: raw trace; orange: median-filtered trace; green: worm-like chain model fit up to the unfolding event. Inset: the extension after subtracting the extension of DNA linker showed a clear unfolding step of ~15 nm. The corresponding region of the force-extension curve is marked by a rectangle. **b** Rupture forces from the taxol-MT pulling traces were significantly lower than the ones from the surface-tethered DNA linker controls ($n$ = 48 and 37 events for taxol-MT and control traces respectively; $p$ = 0.0012, two-tailed Mann-Whitney $U$ test). Error bars: median, 25th and 75th percentiles. Note that the average rupture forces measured from the linker-only control were likely to be an underestimation because the measurements were capped by the maximum force that could be generated by our optical trap (~55 pN; dashed line in Fig. 3B). **c** Rupture-force histogram in the taxol-stabilized microtubule pulling experiments ($n$ = 48 rupture events).

chain model, which included an elasticity term (Fig. 3a, green curve)[36,37]. We observed both stepwise lengthening and ruptures in the pulling traces (Fig. 3a, arrows). After the rupture events, the polystyrene microsphere diffused away after the laser trap was turned off, showing the bead was no longer tethered to the microtubule. Stepwise-lengthening events were often observed prior to the rupture (Fig. 3a, 44 out of 48 traces with rupture events). The lengthening events were absent in control force-extension traces in which DNA-linkers were tethered directly to the surface (zero out of 37 traces; see example of a control DNA-linker pulling trace in Supplementary Figure 4). We therefore interpreted stepwise lengthening as partial unfolding of tubulin subunits.

The rupture events occurring at forces 33.8±1.2pN, (mean ± SE unless otherwise noted, $n$ = 48 traces; Fig. 3b points to the left)

are primarily due to removing tubulin from the microtubule lattice, based on the following arguments. First, breaking the covalent bond between the oligonucleotide and the tubulin protein is expected to require much higher forces (on the nanonewton scale[38]). Second, the shear force to break the 37-nucleotide oligoDNA-DNA linker interaction is expected to be ~60 pN[39], much higher than the measured rupture forces. And third, the force to break the antibody-digoxigenin bond must be larger than the force required to rupture the surface-tethered DNA bead controls, which also used the same anti-digoxigenin antibody (Fig. 3b points to the right; 40.3±1.8pN; $n = 37$ traces). Thus, a great portion of the rupture events observed in the taxol-microtubule pulling experiments likely corresponds to the removal of tubulin by the applied force.

From the distribution of rupture forces (Fig. 3c), we estimated that the force typically required to remove tubulin from the taxol-stabilized microtubule lattice to be around 30 pN at this pulling velocity (~0.32 μm/s). A complication of our optical trap assay is the uncertainty of the actual pulling direction with respect to the tubulin being pulled as we were unable to identify the protofilament on which the subunit was located. The force required to extract tubulin when pulling in the optimal direction (perhaps orthogonal to the lattice surface) is therefore likely to be lower than the rupture forces measured in our optical tweezer assay. We return to this issue in the Discussion.

**Unfolding of tubulin subunits under mechanical force**. To estimate the unfolding step size, we subtracted the length of the DNA linker estimated by fitting the worm-like chain model (Fig. 3a, green curve). The net extension showed clear stepwise increases, which we interpret as the partial unfolding of tubulin (Fig. 3a, inset). The typical step size was 10–20 nm (3 to 92% range) (Fig. 4a), equivalent to the unfolding of ~25–50 amino acids of the polypeptide (~0.4 nm/residue[40],). Examining the secondary structure of α-tubulin close to the C-terminus[41], we hypothesize that these events corresponded to the unfolding of the last C-terminal 2 to 3 helices of α-tubulin (H11, H11', H12, blue in Fig. 4b); this is consistent with an earlier coarse-grained molecular simulation of α-tubulin pulling trajectories[42].

To obtain more detailed kinetic information about the unfolding events, we transformed the unfolding forces associated with the 10-20 nm step sizes (Fig. 4c) into the force-dependent lifetime of the folded state $\tau(F)$ (Fig. 4d) using the Dudko method of[43]. Though the pulling axis in our optical trap assay is likely to be a poor reaction coordinate, the force-dependent lifetime was nevertheless well-described by Bell's model $\tau(F) = \tau_0\exp(-(Fx^{\ddagger})/(kBT))$ (ref [44]), with the lifetime of the folded state at zero force $\tau_0 = 25.9 \pm 1.6$ (s) and the distance to the transition state $x^{\ddagger} = 1.05 \pm 0.10$nm (Fig. 4d). Thus, these results suggest that the unfolding of the region spanning H11 to H12 of α-tubulin is an intermediate step in the tubulin-removal pathway, and that the spontaneous unfolding of this region may take place on the timescale of tens of seconds (but see last paragraph of the Results section).

**Tubulin extraction from GDP-microtubules by molecular motors**. Because the mean force required to extract tubulin with the optical tweezers (34 pN) is not so different from the force to rupture the DNA-bead tether (41 pN), we sought an alternative method to measure the tubulin-removal force. In addition, we sought a high-throughput assay, unlike the tweezer assay in which we can only pull on one tubulin at a time. We therefore developed a motor-pulling assay in which many force events can be measured simultaneously[45]. To this end, we first prepared GDP-microtubules capped with GMP-CPP tubulin (GMP-CPP is a slowly hydrolysable GTP analogue) to prevent their spontaneous

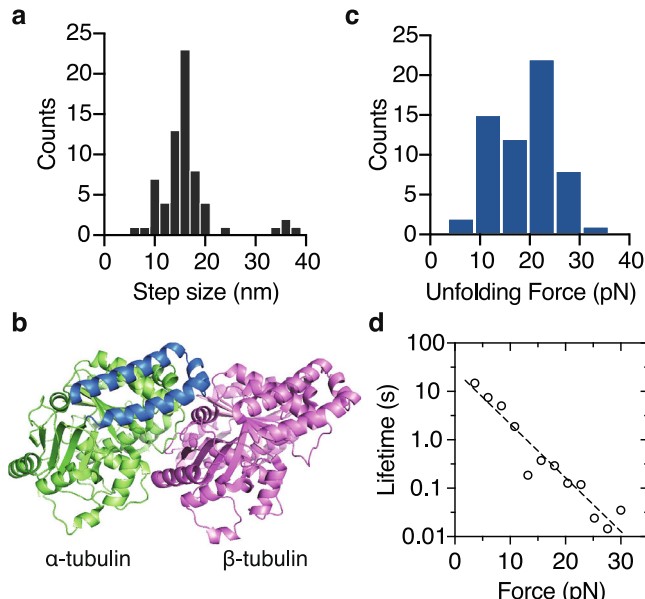

**Fig. 4 Characterization of the unfolding events of tubulin during the microtubule-pulling experiments. a** The distribution of the unfolding step sizes (16 ± 6 nm; mean ± SD, $n = 66$ unfolding events). **b** Structure of αβ-tubulin looking at the microtubule surface (PDB: 3J6F[76]). The hypothesized unfolding 50 amino acids of α-tubulin is highlighted in blue (Helix 11, 11', 12). **c** Distribution of unfolding forces associated with step sizes between 10 and 20 nm (19.2 ± 0.8 pN; mean ± SE, $n = 60$ events). **d** Lifetime of the folded state $\tau$ as a function of force $F$ transformed from the unfolding force histogram in **c** (circles). Fit with the Bell equation (dashed line): $\tau_0 = 25.9 \pm 1.6$ s, $x^{\ddagger} = 1.05 \pm 0.10$ nm (SE); $R^2 = 0.92$.

depolymerization. The GDP-microtubules were prepared by copolymerizing tubulin conjugated to biotinylated DNA linkers (3.8 kb) along with digoxigenin-labeled tubulin in the presence of GTP, which subsequently hydrolyzes to GDP (see Materials and Methods for details). These microtubules were bound to the surface by antidigoxigenin antibody (Fig. 5a). Biotinylated kinesin-1 (a truncated rat kif5C) was then joined to the DNA linkers via neutravidin (Fig. 5a). Note that depending on the concentration, the number of kinesin-1 motors per DNA linker varies from 1 to 3 (there are four biotin-binding sites per neutravidin tetramer with one used to attach the DNA linker). We stained the DNA linkers with the fluorescent DNA intercalator SYTOX Green and visualized them using TIRF microscopy. In the presence of ATP, the DNA linkers, which formed a compact random coil of diameter ~0.5 μm, were quickly stretched along the microtubule filament by the kinesins (Fig. 5b,c, Supplementary Movie 1). The fluorescence intensity of the DNA linker increased as it approached its contour length due to the tension-dependent enhancement of affinity to SYTOX green[46,47]. Thus, the stretching of the tubulin-DNA complexes by the motors could be directly visualized.

After the DNA linker was fully stretched, the motors stalled for a variable time until one of three events occurred. (i) The stretched DNA recoiled to the anchor point and remained attached to the microtubule (yellow arrowheads in Fig. 5b), presumably because the motors dissociated from the microtubule lattice. (ii) The stretched DNA separated into two parts, one part recoiling to the anchor point on the microtubule and the other part recoiling and moving along the microtubule (see an example in Supplementary Figure 5a). We attribute these events to the photocleavage of the DNA. (iii) The stretched DNA recoiled and was transported along the microtubule filament by the kinesins

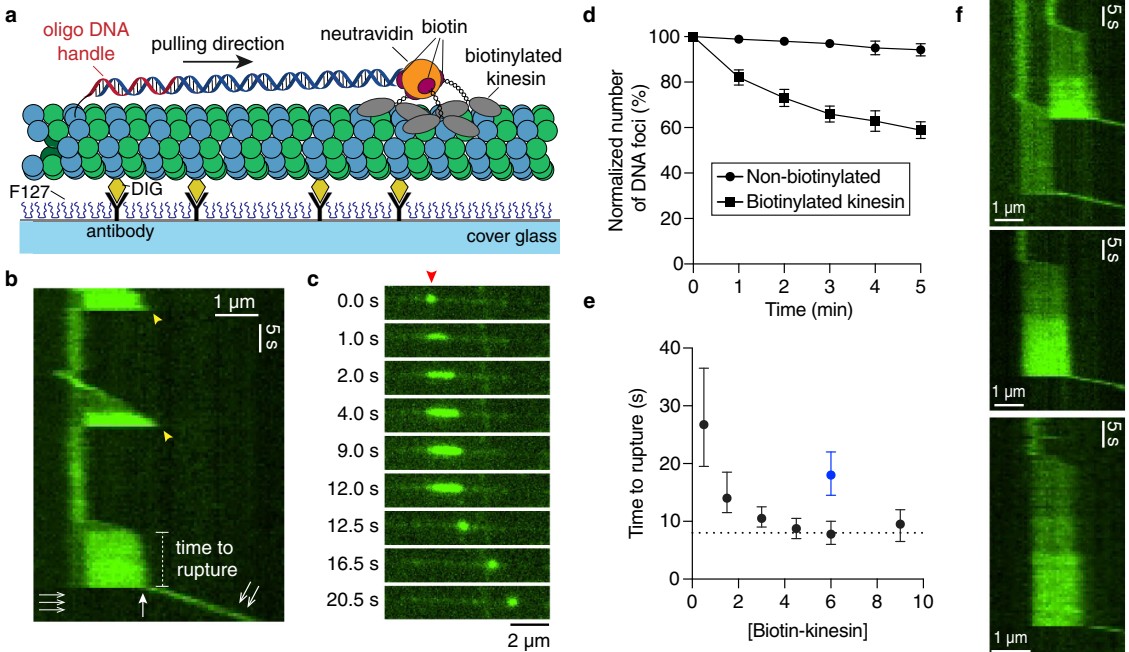

**Fig. 5 Kinesin-1 pulled out tubulin subunits from the GDP-microtubule lattice. a** Experimental scheme of the motor-pulling assay. GDP-tubulin subunit attached to a DNA linker (3.8 kb) was pulled by biotinylated kinesin-1 molecules along the microtubule axis. The GDP-microtubules were stabilized with GMPCPP-tubulin caps on both ends (not shown for simplicity). **b** Example kymograph of kinesin-1 stretching a DNA-linker (stained by SYTOX Green) and pulling out a GDP-tubulin subunit. The dissociation of kinesin-1 is indicated with yellow arrowheads. The rupture event corresponding to the dissociation of tubulin was marked by the white arrow. **c** Time-lapse images of the last stretching event before rupture from the kymograph in **b**. The DNA molecule of interest is indicated with a red arrowhead. **d** The number of microtubule-linked DNA molecules remaining after introducing 6 nM of kinesin-1 (either non-biotinylated or biotinylated with ~50% stoichiometry) imaged at 1 frame per minute (error bars: SDs). Biotinylated kinesin-1 led to a significant reduction of fluorescent DNA molecules after 5 min as compared to the non-biotinylated kinesin-1 control (two-tailed Welch $t$-test $p = 0.0003$; percentage of DNA remained: 59 ± 4% vs. 94 ± 3%; mean ± SD, $n = 3$ experiments). **e** Time-to-rupture (error bars: median ± 95% confidence interval) decreased with increasing concentration of biotinylated kinesin-1 to a value of ~8 s (dashed line). Data collected from triplicate experiments with $n = 108, 105, 130, 122, 112, 99$ events at each concentration (from 0.5 nM to 9 nM). Rupture time from taxol-stabilized microtubules is shown in blue ($n = 160$ events from triplicates). **f** Examples of stepwise increase in fluorescence intensity corresponding to the stepwise increase in force due to the increase in the number of motors engaged in force generation (Supplementary Movie 2 corresponding to the kymograph on the top). An additional example is included in Supplementary Figure 5b,c.

(double arrow in Fig. 5b). Note that no DNA remained associated with the anchor point (triple arrow in Fig. 5b). We attribute these events to the removal of the GDP-tubulin subunit from the lattice by the kinesins (Fig. 5b white arrow, 5c).

As further evidence that the motors are actively pulling the linker DNA off the microtubule lattice (presumably with the tubulin attached to it), we quantified the number of DNA molecules remaining on the microtubules over time by imaging at low frequency (snapshots with 1 min interval) to minimize photobleaching. After introducing biotinylated kinesin-1 and ATP, the number of DNA molecules conjugated to the microtubules decreased by ~40% in 5 min, while more than 90% of the DNAs remained when the same concentration of non-biotinylated kinesin-1 (and ATP) was used (Fig. 5d). This decrease was not a result of tension-enhanced photocleavage because most of the DNA molecules were in the coiled state. This result demonstrates that DNA is removed by biotinylated motors and not by photocleavage.

We measured the time-to-rupture at different concentrations of biotinylated kinesin. The median rupture time decreased with increasing kinesin concentration and approached a plateau of ~8 s (Fig. 5e). The decreased rupture time at higher kinesin concentrations is consistent with more kinesins pulling in the higher concentration assays. The stepwise increases of fluorescence at high motor densities (Fig. 5f, Supplementary Figure 5b, c, Supplementary Movie 2), suggest that up to three kinesins can

bind the neutravidin linker and cause a tension-dependent increase in the fluorescence of SYTOX green. The plateau of 8 s is consistent with our finding that the rupture-time histogram at saturating concentrations of biotinylated kinesin (4.5–9 nM) was well fit by an exponential with a decay time of 8.8 s (Supplementary Figure 6a). The attachment time of ~30 s at low kinesin concentration (Fig. 5e) is much longer than typical unloaded run times of kinesin and the run times measured in optical trapping assays; this supports the notion that kinesin has a catch-like association with the microtubule in which the lifetime of the attached state increases with load along the axis of the microtubule[48,49].

**Comparison of the optical tweezer and motor pulling assays**. To compare the optical-tweezer and motor assays using the same type of microtubules, we measured the rupture time for taxol microtubules using a saturating concentration of biotinylated kinesin (6 nM). We found that the median rupture time in the presence of 10 μM taxol (~18 s; Fig. 5e blue point and Supplementary Figure 6b,c) was longer than the rupture time of non-stabilized GDP-microtubules (~8 s; Fig. 5e, Supplementary Figure 6c), consistent with taxol stabilizing tubulin in microtubules. Interestingly, while taxol decreases the depolymerization rate by ~1000-fold, the rupture time in our motor assay only increased by about two-fold.

To compare the lifetimes of tubulin in the motor assay, in the optical tweezer assay and in the absence of force, we considered a two-step tubulin extraction model based on our finding that tubulin partially unfolds before it is irreversibly removed from the lattice. We assume that the partially unfolded intermediate can rapidly refold in the absence of force but is inhibited when the external force is applied, as is often found in unfolding experiments[50] (Supplementary Figure 9 and Appendix in Supplementary Information). The presence of a reversible unfolding step prior to the tubulin extraction provides an explanation for why the spontaneous dissociation rate of lattice-embedded tubulin in the absence of force is very low (lifetime expected to be ~$10^6$ s, corresponding to one tubulin dimer per micrometer of microtubule per 10 min), and much lower than the ~40 s estimated from the rupture force histogram (Supplementary Figure 7) using the Dudko method[43], which assume a single-step extraction. The unfolded intermediate also reconciles the rupture times in the motor-pulling assays (8–18 s, Fig. 5e) with the lifetime expected from the optical tweezer experiments, assuming that the maximum force in the motor assays is ~10 pN, corresponding to the maximum force generated by 1–2 kinesins (one kinesin generates 4–7 pN[8]). Thus, the rupture forces measured in our optical tweezer assay are consistent with the lifetimes of tubulin in the motor assays as well as the expected long lifetimes in the absence of force.

## Discussion

Using single-molecule techniques, we have shown that tens of piconewtons of mechanical force can partially unfold and remove tubulin subunits from the microtubule lattice on the timescale of seconds. Because the rupture forces in the optical tweezer assay and the rupture times in the motor assays are consistent with each other, as shown in the last section in the Results, experimental limitations of the assays and details that differ between the assays are unlikely to have drastic effects on the results. The limitations of the tweezer assay include the possibility that some ruptures in the tweezer experiments are due to breaking the antibody-digoxigenin bond, and that the reaction pathway for tubulin extraction may be more complex than a two-step process. Differences between the two assays include the surface attachments of the microtubules, the tethering of the DNA linker to the beads or motors, and the directions of the forces. Thus, using two different assays gives additional confidence to the results.

To compare our measured rupture force of 30 pN in the tweezer assay to the forces generated by severases, walking motors and microtubule bending, we need to consider both the directions of the forces and their attachment points to tubulin. The first important consideration is the direction of the force. In our assays, the forces are approximately parallel to the coverglass surface: perpendicular to the axis of the microtubule in the case of the tweezer assay (Fig. 2), and parallel to the axis in the motor assay. While the motor assay is likely to have the same direction independent of the circumferential location of the tubulin, the tweezer assay will pull tangentially to the microtubule surface on tubulins located on the top surface of the microtubule, but more radially (i.e., orthogonally) on tubulins located on the side towards the optical trap. While it is expected that the rupture force depends on the 3-dimensional direction of the force, the fact that there is only a moderate variation of the force in the tweezer assay (SD/mean = 24%), and that the forces in the tweezer and motor assays are consistent with each other (see Appendix in Supplementary Information), implies that the extraction forces do not depend drastically on the direction of the force (when tubulin is pulled by its tail).

Severases are thought to pull tubulin by its tail perpendicular to the microtubule surface[11]. Structural studies suggest that AAA unfoldases, such as ClpXP, ClpB, Vps4 and spastin/katanin, translocate 2 amino acids (~0.8 nm) through their central pores per ATP hydrolyzed[21,23,25]. If these enzymes are well coupled under load, they could potentially generate forces as high as 120 pN ($\Delta G_{ATP}$/0.8 nm), sufficient to rapidly unfold and remove tubulin from the microtubule lattice. Indeed, ClpXP[19,20] and ClpB[18] can generate forces up to 20 and 50 pN respectively, sufficient to partially unfold and extract tubulin. While the force generated by the microtubule severases remains unknown, our force spectroscopy of tubulin supports the plausibility of spastin and katanin using an unfoldase-type mechanism where the mechanical force exerted on the tubulin C-terminal tail threads tubulin through the pore, partially unfolds it, and then pulls the subunit from the lattice[11,51].

Freely walking kinesin and dynein motors proteins remove tubulins from GDP microtubules[27–29]. When these motors walk along microtubules, they generate axial forces with similar orientations to the forces in our motor assay. Motors can generate axial forces up to 4–7 pN when stalled by external loads[8], and likely have force components that are orthogonal to the microtubule surface[52,53]. However, the forces in unloaded assays, where free motors are walking at high speed, are much smaller, <1 pN, a lot less than the stall force[54]. This low force likely indicates that states in which both heads are attached and generating high intramolecular forces are short-lived[55]. How frequently will a tubulin extraction event occur when a single molecular motor walks on the microtubule track? Assuming our motor pulling assay can serve as an approximation of the tubulin extraction process during a motor's walk, we expect a tubulin extraction event to take at least 30 s when the stall force (4–7 pN) of a single kinesin is applied (based on the rupture time measured in the presence of lower concentration of biotinylated kinesin-1, Fig. 5e, 0.5 nM). The time is likely to be much longer if the force is only 1 pN. Given that the kinesin run-time along the microtubule is ~1 s, we expect that « 1% of kinesin runs will result in tubulin removal. Therefore, a large fraction of the GDP-microtubules are expected to survive for several minutes before they depolymerize, even in the presence of high motor concentrations, as observed[27]. It is also important to recognize that kinesins and dyneins interact with helix 12 on α and β-subunits[56,57], distinct from the α-tubulin CTT that serves an anchor point in our assay, and so the lifetimes in the freely walking assays could be different (larger or smaller) to the lifetimes in our tethered-motor assays.

Microtubule bending also destabilizes microtubules by leading to tubulin removal from the lattice. A radius of curvature of a few micrometers is expected to generate tensile forces up to 100 pN in the outer protofilaments of a microtubule (assuming a Young's modulus of 2 GPa[8]), and compressive forces of similar magnitude in the inner protofilaments. This force is larger than the rupture forces measured in our assays. But the anchor points of the tensile and compressive forces correspond to the interface between the dimers, quite different to the C-terminal tail. Thus, when comparing the magnitude of forces required to extract tubulin from the lattice in our assays to other assays, it is necessary to consider both the anchor point on the dimer on which the force acts, as well as the 3-dimensional direction of the force.

Our motor-pulling assay demonstrates that the forces generated by a few kinesin-1 molecules are sufficient to extract GDP-tubulin from the lattice. Removal of tubulin from the unstabilized GDP-tubulin lattice did not lead to immediate microtubule depolymerization (at least within minutes), however. This suggests that the lattice destruction observed when dyneins and kinesin-14s walk along microtubules, and when microtubules glide across surfaces coated with these motors may be due to accumulated damage to the lattice by several motors[27]. For cellular processes such as mitosis[9] and tissue development[58] that

involve frequent sliding of microtubule filaments by multiple motors, the removal of tubulin under force may create defects on the microtubules and require repair by reincorporation of new GTP-tubulin to the lattice to prevent filament breakage[31].

In conclusion, our results provide the first experimental investigation of the mechanical force required to unfold and extract tubulin from the microtubule lattice. Several important questions remain to be explored. For instance, while we hypothesize one single tubulin dimer is removed based on the strong binding between α- and β-tubulin, it is not known whether the adjacent tubulin subunits also dissociate from the lattice. Does the partial unfolding of tubulin weaken tubulin-tubulin bonds, facilitating the dissociation of the subunits? How does the unfolding and rupture force depend on the pulling orientation? Is there a difference in force requirement when pulled on the C-terminal tail or helix 12 of β-tubulin as compared to α-tubulin? We expect our assay design to provide an important step toward understanding the molecular response of tubulin to mechanical force, with important implications in the mechanics of the microtubule cytoskeleton and its cellular functions.

## Methods

**Protein preparation and assays**. Bovine brain tubulin was purified in house based on the method described previously[59]. Codon-optimized human tubulin-tyrosine ligase (TTL) was custom synthesized (IDT) and cloned into a pET vector with N-terminal His$_6$-SUMO tag (Addgene plasmid #29659; a gift from Scott Gardia, University of California Berkeley) by ligation-independent cloning. His$_6$-SUMO-TTL was expressed in *Escherichia coli* (BL21-DE3 competent cells, Agilent) overnight at 18 °C in LB broth (induced by 0.5 mM IPTG). The cells were harvested and stored at -80 °C until purification. To purify His$_6$-SUMO-TTL, the cells were resuspended in cold lysis buffer (20 mM NaH$_2$PO$_4$, 0.2 M NaCl, 5 mM imidazole, pH 6.9, 0.2 mM pefabloc, 0.1 mg/mL lysozyme, 1 mM dithiothreitol (DTT), 0.3 U/µL benzonase), and lysed by sonication on ice. The lysate was clarified by centrifugation and loaded onto a HisTrap column (GE Healthcare). The column was washed with the imidazole buffer (20 mM NaH$_2$PO$_4$, 0.2 M NaCl, 20 mM imidazole, pH 6.9, 1 mM DTT). The protein was then eluted with a continuous gradient (20 mM to 500 mM imidazole). The eluted protein was concentrated by centrifugal filters (30 kD cutoff) and further purified by size exclusion chromatography (SEC). The protein was eluted from size exclusion column with SEC buffer (20 mM MES-KOH, 0.1 M KCl, 10 mM MgCl$_2$, 1 mM EGTA, 1 mM DTT, pH 6.9). The purified protein was concentrated with centrifugal filters, aliquoted, flash frozen with liquid nitrogen and stored at -80 °C.

To test the tyrosination activity of recombinant TTL, 1 µM of purified His$_6$-SUMO-TTL was incubated with 10 µM of bovine tubulin in the presence of 2.5 mM ATP, 1 mM L-tyrosine and 5 mM DTT in BRB80 (80 mM PIPES-KOH, 1 mM EGTA, 1 mM MgCl$_2$, pH 6.9) at 37 °C. A negative control without tyrosinated tubulin was prepared by treating 3 mg/mL bovine tubulin with 0.02 mg/mL carboxypeptidase A (CPA, Sigma Aldrich) on ice for 15 min to proteolytically cleaved the C-terminal tyrosine of α-tubulin. The tyrosinated tubulin level was detected by Western blot using monoclonal anti-tyrosinated tubulin antibody (clone YL1/2; 1:2500 dilution; EMD Millipore) as the primary antibody and a rabbit anti-rat secondary antibody (alkaline phosphatase-conjugated, 1:1500; Invitrogen). α-tubulin was detected by Western blot using antialpha tubulin antibody (clone DMA1A; 1:5000; EMD Millipore) along with an alkaline phosphatase-conjugated goat anti-mouse secondary antibody (polyclonal, Invitrogen, 1:5000 dilution).

For the motor-pulling assay, we used a truncated rat kinesin-1 (the first 430 aa of rat Kif5C, here denoted as rk430)[60] fused with a mScarlet-SNAP-His$_6$ tag at the C-terminus. The expression of rk430-mScarlet-SNAP-His$_6$ was performed in the same way as TTL as described earlier. The cell pellet was resuspended in cold lysis buffer (30 mM HEPES-NaOH pH 7.4, 10 mM imidazole, 0.3 M NaCl, 1 mM DTT, 10 µM ATP) containing protease inhibitor (Roche) and 0.3 mg/mL lysozyme. The cells were then lysed by sonication and clarified as earlier described. The clarified lysate was flowed through a HisTrap column and eluted with a gradient of imidazole (from 10 to 500 mM). The peak fractions were collected, desalted with Zeba desalting column (Thermofisher) into anion-exchange buffer (30 mM HEPES-NaOH pH 7.4, 1 mM DTT, 10 µM ATP) and bound to a HiTrapQ column (GE Healthcare). The proteins were eluted with a NaCl gradient (0.01 to 1 M). The peak fractions were collected, concentrated using an Amicon centrifugal filter and purified further with SEC, eluting with BRB80 containing 10 µM ATP and 0.1% β-mercaptoethanol. The purified protein was flash frozen with liquid nitrogen as small aliquots and stored at -80 °C. The activity of purified kinesin was confirmed by the single-molecule fluorescence stepping assay as previously described[61].

**Site-specific labeling by tubulin-tyrosine ligase (TTL)**. To introduce the bio-orthogonal azide functional group, bovine brain tubulin was mixed with His$_6$-SUMO-TTL (5:1 molar ratio of tubulin: TTL) in the presence of 1 mM 3-azido-tyrosine (Watanabe Chemical), 2.5 mM MgATP in BRB80 and incubated at 37 °C for 45 to 60 min. To remove TTL and excessed azido-tyrosine, the labeled tubulin was polymerized into microtubules by adding GTP (final concentration 1 mM), glycerol (final concentration 33.3%) and MgCl$_2$ (final concentration 3.5 mM) at 37 °C for 30 min. The microtubule solution was layered on a glycerol cushion (BRB80 with 4 mM MgCl$_2$, 1 mM GTP, 60% glycerol) and centrifuged (340,000 x g, 35 min at 35 °C). The pellet was resuspended in warm click-labeling buffer (BRB80, 1 mM GTP, 4 mM MgCl$_2$, 40% glycerol), followed by the addition of oligo-DNA or small molecules conjugated to dibenzocyclootyne (DBCO) (~0.5 mM final concentration for organic fluorophores (ThermoFisher) or biotin with PEG$_4$ linker (Sigma Aldrich); or ~0.15 mM oligo DNA: 5'-TGGACTGATGCGGTATCTGCGA-TATCCTACGCAGGCGTTT-3'-DBCO (synthesized from IDT). The copper-free click reaction took place at 37 °C for 1 h with gentle shaking and occasional mixing. The labeled microtubules were then again layered on the 60% glycerol cushion and centrifuged at 35 °C (446,000 x g, 20 min) to remove the unreacted labeling reagent. The microtubules were resuspended with BRB80 and incubated on ice for 30 min to induce depolymerization. The solution was centrifuged at 4 °C (285,500 x g, 10 min) to remove aggregates or precipitates. An additional polymerization and depolymerization cycle was used to further purify functional tubulin based on a previous protocol[62]. The labeled tubulin was flash frozen in liquid nitrogen and stored at -80 °C. The concentration of tubulin and labeling density of fluorophores was determined by UV-Vis absorption. Biotinylation level was quantified using a biotin quantification kit (Thermofisher). The oligo-DNA labeling density was estimated by measuring the band intensities from SDS-PAGE. Note that we chose copper-free click chemistry due to the commercial availability of azido-tyrosine and its faster reaction kinetic without the need of catalyst as compared to the hydrazone formation strategy used in[33].

To confirm the labeling specificity, high-resolution SDS-PAGE was used to separate α-and β-tubulin as previously described[63]. To examine the specific labeling site, SSL-biotinylated tubulin was first buffer exchanged into phosphate buffer saline (PBS) pH 7.4. Biotinylated tubulin was then protease digested by AspN (Promega) at 37 °C overnight (1:50 w/w ratio). An equal volume of BRB80 was added to stop the protease digestion and incubated with NeutrAvidin agarose resin (Pierce) for 1 h at room temperature to pull down the biotinylated peptides. The resin was washed with PBS and water, and the biotinylated peptides were eluted by incubating the resin with an aqueous solution containing 80% acetonitrile, 0.2% trifluoroacetic acid, 0.1% formic acid at 60 °C for 5 min. The elute was then sent for proteomic analysis. The peptides were separated using reverse-phase ultra-high pressure liquid chromatography (UPLC) and analyzed by tandem mass spectrometry (MS/MS) with electrospray ionization. MS/MS spectra were searched by MASCOT against the bovine α-tubulin sequence with the modifications corresponding to the copper-free click biotin adduct on tyrosine and polyglutamylation (up to 4 glutamate chains) on glutamate side chains. All biotinylated peptide hits corresponded to the C-terminal tail of α-tubulin, confirming the site-specificity of the labeling method. Microtubule dynamic assays were used to measure the dynamic properties of SSL-Alexa Fluor 488-tubulin and unlabeled tubulin (in BRB80 supplemented with 1 mM GTP and 5 mM DTT) as described previously[64,65] using interference-reflection microscopy (IRM)[66].

**Microscopy setups**. The two-color (488 nm and 561 nm excitation lasers) total-internal-reflection-fluorescence (TIRF) microscopy and interference-reflection microscopy (IRM) was set up on a Nikon Ti-Eclipse microscope as previously described[66] with a sCMOS camera (Zyla 4.2 Plus, Andor) for both TIRF and IRM imaging. Microscopy images were acquired by using Nikon NIS elements.

The home-built optical tweezer setup was constructed around a Zeiss inverted microscope as previously described[67] with a near infrared 1064 nm Nd:YAG laser (IPG photonics). The system was coupled to a differential-interference-contrast (DIC) condenser to visualize microtubule filaments and the polystyrene microspheres using blue-light LED illumination[68]. We used a Zeiss Plan-Neofluar 100x/1.3 NA objective for both trapping and DIC imaging. The sample stage was controlled by a three-axis piezoelectric stage with sub-nanometer precision (Physik Instrumente). To detect the position of the bead with nanometer sensitivity, we used a quadrant photodiode (QPD; First Sensor) for back-focal-plane detection and calibrated the position detection and trap stiffness using a combined drag force-power spectral analysis method as described[67,69]. All optical tweezer experiments were performed at similar trap stiffness (~0.28-0.32 pN/nm) and the time traces were recorded at 1 kHz sampling frequency. The optical tweezer system was operated by custom-written software using LabView (National Instruments)[67].

**Preparation of microspheres**. The carboxylated polystyrene microspheres (0.58 µm diameter; Bangs Laboratory) were labeled with anti-digoxigenin Fab fragments (Roche) based on a two-step functionalization method described in[70] with the following modifications. First, we used a 3:1 molar ratio of 2 kDa α-methoxy-ω-amino PEG: 3 kDa α-amino-ω-carboxy PEG (Rapp Polymere) in the first functionalization step. Second, we used 0.16 mg/mL anti-digoxigenin Fab in phosphate buffer saline (PBS) pH 7.4 for the antibody conjugation step. The Fab-conjugated microspheres were sonicated briefly (~15 s) and incubated with 1 mg/mL BSA on ice for at least 10 min prior to the optical tweezer experiments to enhance the bead surface passivation.

**Preparation of DNA linkers**. The long DNA linkers containing a 5' overhang on one end and digoxigenin on the other end were prepared by PCR[71,72] using Q5 polymerase (NEB). Lambda phage DNA (Thermofisher) was used as the PCR template. The digoxigenin-labeled primer was prepared by labeling 0.25 mM oligo containing 3 amino groups (amino groups at the 5' end and the underlined bases: TCTAAGTGACGGCTGCATACTAACC; synthesized by IDT) with 66 mM digoxigenin N-hydroxysuccinimide ester (Roche) in 50 mM HEPES-NaOH, 67% DMSO, pH 8.3 overnight with shaking at room temperature. The labeled primer was purified by G-25 microspin column (Cytvia) and Monarch DNA purification kit (NEB). The successful conjugation of three digoxigenin moieties was confirmed by the shift of the molecular weight using denaturing urea polyacrylamide gel electrophoresis. The primer with 5'- overhang was custom synthesized (IDT) with the following sequence: CGCCTGCGTAGGATATCGCAGA-TACCGCATCAGTCCAXCAACGGTCGATTGCCTGACGGA where the sequence complementary to the oligo handle labeled on tubulin is underlined and X is the abasic residue (1',2'-dideoxyribose). This primer pair produced the 8.2 kb DNA linker used for the optical tweezer taxol-stabilized microtubule pulling assay. For surface-tethered DNA control, the purified 8.2 kb DNA linker was annealed to the oligo DNA handle with 3'-biotin (5'-TGGACTGATGCGGTATCTGCGA-TATCCTA CGCAGGCGTTT-3'-biotin) in PBS at room temperature for 3 hr followed by purification using Monarch DNA purification kit (NEB).

For the motor pulling assay, the digoxigenin-labeled primer was replaced with 5'-biotinylated primer (IDT). The complete primer pairs and the corresponding product length is provided in Supplementary Table 1. The DNA linkers produced by the PCR reaction were first desalted using an amicon centrifugal filter and purified using the PureLink PCR purification kit (Thermofisher). DNA linkers were eluted with low TE buffer (10 mM Tris-HCl, 0.1 mM EDTA, pH 8.0) and stored at -20 °C as small aliquots.

**Taxol-stabilized microtubule pulling assay with optical tweezer**. To prepare taxol-stabilized microtubules conjugated to DNA linkers, the tubulin-DNA mixture with 20 μM of tubulin labeled with oligo DNA (labeling density ~3%), 4 μM of biotinylated tubulin (labeling density ~50%), ~3.4 nM 8.2-kb DNA linker was incubated on ice for 15 min and polymerized at 37 °C in the presence of 4 mM MgCl$_2$, 2% DMSO, 1 mM GTP for 25 min. The polymerized DNA-microtubules were quickly diluted into BRB80 supplemented with 10 μM taxol and pelleted by centrifugation. DNA-microtubules were then resuspended in BRB80 containing 10 μM taxol.

The flow channel was prepared by using one 18×18 and one 22×22 mm$^2$ Piranha solution-cleaned and silanized coverslips sandwiching parafilm stripes as previously described[65,73]. The flow channel was incubated with 0.02 mg/mL biotinylated bovine serum albumin (BSA) (Sigma Aldrich) solution for 5 min and washed by BRB80. The channel was then passivated by 3% BSA solution for 30 min and washed with BRB80 + 1%Tween20. The chamber was further incubated with 0.05 mg/mL Neutravidin solution containing 1 mg/mL BSA and 1% Tween20 for 10 min and washed again with BRB80 + 1%Tween20. Taxol-stabilized DNA-microtubules were then attached to the surface under solution flow. Polystyrene beads conjugated with anti-digoxigenin Fab (~0.01% beads in BRB80 supplemented with 0.1%Tween20, 1 mg/mL BSA and 10 μM taxol) were then introduced to the chamber and incubated for 30 min at room temperature to allow the attachment of DNA linkers. The unbound beads were then washed out. Freshly prepared oxygen scavenger solution [40 mM glucose, 0.04 mg/mL glucose oxidase (from *Aspergillus niger*; Sigma Aldrich), 0.02 mg/mL catalase (from *Aspergillus niger;* EMD Millipore), 0.2 mg/mL casein, 10 mM DTT, 10 μM taxol, 0.1% Tween 20] was then introduced into the channel. The channel was then sealed with VALAP (equal ratio of Vaseline, lanolin and paraffin) for the optical tweezer experiments. Tethered beads undergoing tethered Brownian motion close to the microtubules (imaged by DIC) were selected for optical trapping and the calibration was performed for each bead. The tethered bead was trapped at ~300 nm away from the surface and the stage was first set to oscillate with a small amplitude (2.5 to 2.8 μm) in the lateral direction of the microtubule at 0.1 to 0.2 Hz. The position of the stage was finely adjusted so that the trap center was aligned with the anchor point of the linker based on the symmetry of the QPD voltage-distance traces. The amplitude we used in this process was quite small so that the force was typically less than 5 pN. To collect the pulling traces, the stage was oscillated with constant velocity (~0.32 μm/min) with an amplitude of 3.2 or 3.3 μm until rupture was observed (typically within 1 or 2 cycles). More than 95% of the traces showed rupture events in our experiments. All traces were collected within 60-75 min after the oxygen scavenger solution was introduced.

For the surface-tethered DNA control experiments, 22×22 mm$^2$ coverslips were first cleaned by three cycles of sonication in 1 M KOH and ethanol (15 min each) and dried under nitrogen gas. To covalently functionalize the cleaned coverslips, 40 μL of biotinylation solution [1 mg/mL of biotin-PEG5000 silane, 100 mg/mL PEG5000-silane (Nanocs) dissolved in 95% ethanol] was sandwiched by two coverslips and incubated at room temperature for 2 hr inside a sealed wet chamber. The functionalized coverslips were then rinsed with water, dried under nitrogen gas and stored. The flow channel was constructed in the same method as the taxol-microtubule pulling experiments. The channel was passivated with 3% BSA and washed with BRB80 + 1%Tween20. To tether the biotinylated DNA-linker, 10 μg/mL Neutravidin solution containing 1 mg/mL BSA and 1% Tween 20 was then

introduced to the channel followed by another wash of BRB80 + 1% Tween 20. The channel was then incubated with 15 pM of biotinylated DNA linkers for 15 min followed by another wash. The successful hybridization of the DNA linkers was verified by staining of dsDNA (QuantiFluor, Promega). Bead tethering and optical tweezer pulling were performed as the taxol-stabilized microtubule pulling assay described above.

**Analysis of pulling traces**. To obtain the force-extension curves from the pulling traces, we employed the strategy described in[36] to identify the anchor point based on the symmetry of pulling traces during stage oscillation. The extensions were calculated by taking the pulling geometry into account. Due to its long contour length, the stretched DNA linker was approximately parallel to the microtubule axis and the vertical component of the force was less than 10% of the axial force. An 11-point moving median filter was applied to the force-extension traces and the traces were fit with the worm-like chain with an elastic term (eWLC) model[36,37](up to the unfolding steps if they were present). Traces that showed evident asymmetry during the stage oscillation cycles or deviated significantly from the predicted DNA force-extension curve were discarded. Rupture forces and unfolding forces were measured from the sharp decrease of the force from the force-extension curves and force-time traces. To estimate the unfolding step size, the DNA extension estimated by the eWLC fit was subtracted from the total extension. The step size was estimated from the jump of the net extension over time. Limited by the noise, the minimum step size we could estimate is on the level of ~5 nm. The unfolding force histogram was converted to the force-dependent folded state lifetime based on the previous work[43] which included the correction for the force-dependent loading rate resulted from the compliance of the DNA linkers. All optical tweezer data analysis was performed using MATLAB software.

**Motor pulling assay**. The flow channel was prepared as aforementioned using 0.01 mg/mL anti-digoxigenin antibody (Roche) and passivated with 1% F127 and 2 mg/mL casein. To prepare the GMPCPP-capped microtubule, digoxigenin-labeled GMPCPP seeds were first polymerized (4 μM tubulin, 0.4 μM digoxigenin-tubulin, 1 mM MgCl$_2$, 1 mM GMPCPP) at 37 °C for 30 min. The GMPCPP seeds were centrifuged and resuspended in BRB80 at a tubulin concentration of ~4 μM (assuming ~70% recovery). The DNA-tubulin mix containing 20 μM of oligo-tubulin (labeling density ~3%), 2 μM of digoxigenin-tubulin, ~8 nM 3.8 kb biotinylated DNA linker, 6 mM MgCl$_2$, 1.5 mM GTP were incubated on ice for 15 min. The DNA-tubulin mix was then quickly warmed up to 37 °C and mixed with the GMPCPP-seeds in a 2:1 volume ratio. The microtubules were then polymerized at 37 °C for 25 min and quickly diluted with 10 times volume of BRB80-taxol solution (BRB80 containing 10 μM taxol). The taxol-stabilized microtubules were centrifuged and resuspended in BRB80-taxol to remove any unpolymerized tubulin and DNA molecules. The microtubules were then introduced into the flow channel; following attachment to the surface, the channel was washed with BRB80-taxol. To perform the capping, taxol was quickly washed out by flowing in BRB80 followed by the GMPCPP-tubulin capping mix (4 μM of unlabeled tubulin, 0.05 mg/mL neutravidin, 5 mM DTT, 0.5 mM GMPCPP, 0.1% Tween20, 2 mg/mL casein). The capping was performed at 28 °C for ~5 min and the channel was washed with BRB80 again. This taxol-washout capping strategy allowed us to bind the GDP-microtubule segment to the surface and ensure that all DNA-tubulin subunits contained GDP rather than GMPCPP. Note that these microtubules should be free of taxol during the motor pulling experiments because the mean unbinding time of taxol is less than 10 s[74], and we also observed occasional shrinkage of GDP-microtubules during the course of our experiments.

To prepare biotinylated kinesin, rk430-mScarlet-SNAP (7 μM monomer) was incubated with benzylguanine-biotin (NEB) in a 4 to 1 molar ratio at room temperature for 15 min, which we found to be sufficient for complete reaction. This labeling ratio reduced the probability that a single kinesin dimer was labeled with two biotin moieties and only half of the kinesin dimers contained biotin (i.e., 50% labeling density). The biotinylated kinesin was stored on ice until the microscopy experiments. To visualize and stretch the DNA-linkers conjugated to GDP-microtubules, various concentrations of biotinylated kinesin were added into the oxygen scavenger solution containing ATP and SYTOX-Green (40 mM glucose, 0.04 mg/mL glucose oxidase, 0.02 mg/mL catalase, 0.2 mg/mL casein, 10 mM DTT, 2 mM Trolox, 0.1%Tween 20, 1 mM ATP, 40 nM SYTOX-Green) and flowed into the flow channel. The DNA molecules were then immediately visualized by TIRF microscopy with a 0.5 s time interval (or 1 min/frame for the photodamage controls). All experiments were performed at 28 °C. For the taxol-microtubule experiments, microtubules were prepared as described in the optical tweezer assay but substituting the biotinylated tubulin with the digoxigenin-labeled tubulin and replacing the 8.2 kb DNA linker with the 3.8 kb biotinylated DNA linker.

All analyses of the motor pulling assay were performed by using Fiji[75]. Rupture time was measured from kymographs. A sharp decrease of stretching velocity typically occurred when the DNA linkers were fully stretched. The rupture time was determined as the time from this velocity transition (with verification by checking whether the DNA was stretched close to its contour length) until the rupture events that corresponded to the removal of GDP-tubulin. Traces that showed clear photocleavage events were excluded from the rupture time measurement.

**Reporting Summary**. Further information on research design is available in the Nature Research Reporting Summary linked to this article.

## Data availability

The raw and source data supporting this study are provided with this paper. The αβ-tubulin structure looking at the microtubule surface referred in this manuscript is publicly available with the PDB accession code 3J6F [https://doi.org/10.2210/pdb3J6F/pdb]. Source data are provided with this paper.

## Code availability

The code used for the data analysis in this study can be found on Github: https://github.com/YinWei-K/OpticalTweezerTraceAnalysis.git.

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

## Acknowledgements

We thank Dr Veikko Geyer, Dr Maijia Liao and Dr Anna Luchniak for the helpful discussion. We thank Dr Ziad Ganim, Dr Ya-Na Chen and Dr Nandan Pandit for their help in the development of optical tweezer assay, Rafael C.A Tavares for the assistance on gel imaging and manuscript writing, and the Yale Keck Proteomics Center for the mass spectrometry analysis. We thank Dr Tom Pollard, Dr Enrique De La Cruz and Dr Yongli Zhang for the feedback on the work. Y.T. acknowledges the support of the Alexander von Humboldt Foundation through the Feodor Lynen Research Fellowship. This work was supported by NIH Grant R01 GM139337 (to J.H.).

## Author contributions

Y.-W.K., M.M. and J.H. conceived the project; Y.-W.K. performed all experiments and data analysis with the assistance of M.M. and J.H.; preliminary data was collected by Y.-W. K. and M.M.; M.M., Y.-W.K. and Y.T. set up the optical tweezer instrument; Y.-W.K. and J.H. wrote the paper with the input of all authors.

## Competing interests

The authors declare no competing interests.
