## [Peer Review File · Nature Communications]

REVIEWER COMMENTS

Reviewer #1 (Remarks to the Author):

The authors present single molecule experiments that investigate the force required to remove a tubulin subunit from the microtubule (MT) lattice. The study is very timely as multiple papers have emerged recently regarding MT lattice damage and repair by motors, and the implications of the study would be of interest of a wide biophysical audience. The authors present two innovative biophysical assays: site-specific labeling of alpha tubulin and the kinesin-DNA-tubulin pulling assay that could be adapted for other studies. Overall, the paper is well-written. However, I have some concerns that should be addressed before publication.

1. Title: The title is very broad. As pointed out throughout the paper, the authors investigate the force to remove tubulin from the MT lattice... but under very specific conditions. As noted in comments below, changing those conditions could have implications on the force required to remove the tubulin subunit. I suggest narrowing the scope of the title to fit the findings in the paper.

2. Abstract: "...AAA-unfoldases, suggesting that severing proteins such as spastin and katanin use an unfoldase mechanism." Maybe, but that statement is not conclusive from this study. I suggest scaling back the language.

3. Abstract: "Our results reveal the response of tubulin to mechanical force and advance..." Very broad. Again, these experiments are under very specific conditions, and a generalized understanding of the force to displace a tubulin subunit from the MT lattice has remaining questions. Narrow the language here.

4. Results, Page 3, line 24: What do the authors mean by a labeling density of 43% Is this saying 43% of all alpha-CTTs are labeled on a MT? Please clarify.

5. Results, Page 4, line 17: The authors use the site-specific labeling (SSL) method to conjugate alpha tubulin CTTs with a single stranded oligo-DNA strand. Controls were performed to show that alpha CTTs labeled with a fluorophore via the SSL method do not have large effects on MT dynamics. But why not do the same control with the oligo-DNA labeled tubulin? From the methods section, it sounds like the oligo-DNA tubulin, biotinylated tubulin, and the longer DNA linker were incubated together before polymerization (Methods, Page 20, starting at line 6). I would imagine that a large, floppy DNA linker could potentially affect MT dynamics. And at what labeling density of the small and large DNA causes differences in MT dynamics? A comparison between the oligo-DNA MT dynamics and full DNA tether-

labeled MT dynamics could also give a clue regarding how those labeled tubulin get incorporated into the lattice. Perhaps local strain due to the large attachment and its Brownian motion “primes” that particular subunit to become displaced in the force pulling experiments.

6. Results, Figure 2 and Page 5, lines 10-11: I’m convinced of the integrity of all of the connections in the pulling assay except for the BSA/coverglass interaction. What forces can the BSA/coverglass interaction undergo before detaching from the surface? Also, how does the relative amount of biotinylated tubulin change the stiffness of the MT? I could imagine that more biotinylated tubulin within the MT lattice would cause more interactions with the neutravidin coated surface (assuming the biotinylated BSA stays put as assumed in the assay) and thus cause a tighter/stiffer interaction of the MT with the coverglass surface. This relative stiffness could also influence interactions within the MT lattice and maybe emphasize any “defects” or “alterations” to the normal lattice pattern that could be influenced by the DNA-labeled tubulin.

7. Results, Page 6, starting at line 4: The authors state that they only do the tweezers experiment by pulling perpendicular to the MT axis. “We chose this pulling direction mainly to avoid the attachment of more than one DNA linker to the bead during the pulling experiment.” While pulling perpendicular is of course interesting, it is possible that pulling direction along the axis of the MT could result in differing rupture force (including differences in plus vs minus end pulling, effects from pulling against the neighboring lattice subunits). Further, perpendicular pulling direction may be relevant for potential mechanisms of severing enzymes, but not necessarily for motor induced damage while walking. In order to pull in different directions could the following assay alterations be implemented to accomplish such pulling?

1. Use fluorescent polarity marked MTs (bright GMPCPP seed, dimmer elongation, taxol stabilized) and dilute the oligo-DNA tubulin concentration such that only one or two tethers can form per MT? That way you can pull either toward the plus or minus end with no interference from other tethers. Yes, that makes an already low throughput experiment more low throughput, but there are multiple MTs per coverglass that could be probed.

2. Or use polarity marked MTs but change the assay order. Incubate the anti-dig beads with the long DNA-dig strand first, then add to the assay with oligo-DNA MTs. You then actively form tethers by trapping the bead, bringing it to the surface, and grazing the surface of the MT until a connection is made. Then pull in the respective direction.

8. Results, Page 6, lines 20-21: It is not clear whether the authors know that a single tubulin subunit is being removed (just alpha tubulin, the whole tubulin dimer, more local damage?). The authors should comment on these possibilities or clarify their language. Perhaps a MT-lattice repair assay with soluble labeled tubulin could help narrow down the level of damage induced.

9. Results, Page 7, lines 3-4: The authors state that a “complication of our optical trap assay is the uncertainty of the actual pulling direction with respect to tubulin...” I thought above the authors were pulling perpendicular to the MT axis? Please clarify.

10. Figure 4A: It appears that there’s a smaller step size distribution at around 35 nm, but these are excluded from unfolding force analysis (page 8, line 23). Why?

11. Figure 5C caption: “Time-lapse images of the last stretching round before rupture...” What does “last stretching round” mean? Do the motors walk to the end of the DNA extension, recoil back after stall, and begin walking again several times (similar to a single molecule kinesin optical trapping assay)? Is it possible that the repeated pulling on the tubulin subunit when the kinesins reach stall help to “pry” the subunit out of place over time?

12. Figure 5: The authors should do a comparison of the kinesin pulling assay with taxol stabilized MTs since the trapping assay uses that form of stabilization.

Reviewer #2 (Remarks to the Author):

Kuo et al Comments

This is important work. The authors use a published method to switch the C-terminal tyrosine of alpha tubulin for a reactive azo-tyrosine, validate its use for site-specific attachment of a DNA linker and microbead to the C-terminus of alpha tubulin in the lattice, and exert calibrated forces on it using either an optical trap or a small team of kinesins. They measure the force required to rupture the connection between the bead and the microtubule, and they show that the force required is (marginally) less than that required to break the chain of linkages when it is connected directly to the substrate, arguing therefore that the observed rupture events indicate tubulins being pulled from the GDP-taxol lattice. They also detect a force dependent unfolding event, likely of the alpha tubulin C-terminal H11-H12 hairpin.

To my knowledge this is the first direct measurement of the force required to extract tubulin from the lattice. There are a number of caveats to each experiment, which the authors carefully consider. The worst case scenario is that some or even all of the ruptures detected actually represent breakage of the chain of connectivity, rather than extraction of the tubulin. But even if this were so, and the extraction force were thereby under-estimated, by setting limits these measurements remain extremely valuable to the field. The experiments are a remarkable technical achievement and the authors have done everything feasible to strengthen the case that the values obtained are correct.

The authors in their discussion acknowledge that much remains to be done (test different types and states of lattice, find out if one molecule or several are being dragged out of the lattice), but this paper contributes much-needed quantification in a currently murky area. I think it will be invaluable.

I have only minor comments.

1. Even in the abstract, the authors try to generalise their result to infer that tubulin-extracting AAA ATPases like katanin could plausibly work by unfolding tubulin and transporting the unfolded chain against a load. This point is worth mentioning but I'm not convinced it belongs in the abstract. At best the current data tell us one way in which katanin might in principle work. But if anything rather than supporting an 'unfold and pump the unfolded chain' model for tubulin extraction, the data say that tubulin extraction involves a preliminary partial (~10% of the primary sequence) unfolding event, after which the tubulin molecule jumps out of the lattice en bloc.

2. The inferred zero force stay-resident lifetime of 40s for GDP-tubulin in the undamaged lattice is hard to believe. If this rate of spontaneous loss of tubulin from the lattice occurs, wouldn't capped GDP microtubules in the absence of free tubulin self destruct? Bending the GDP-tubulin lattice in microfluidic flow is reported to create and expand defects and taxol fully protects. There are modelling-simulation studies that assign definite stabilities (energies) to the lateral and longitudinal bonds stabilising heterodimers in the lattice. Are the energies inferred here consistent?

3. The trapping work is done with GDP taxol microtubules, which as the authors discuss are more stable to mechanical damage than GDP-microtubules. Perhaps make the point therefore that the forces measured in the trapping experiment will tend to over-estimate the force required to extract tubulin from a GDP-only lattice? This is a strength of the approach. A little more detail on why measurements with capped GDP microtubules were intractable would be interesting, since this too may support metastability of strained GDP tubulin molecules in the lattice, which is the point of interest here.

4. Fig S5B&C (stepwise increase in pulling force detected by DNA dye fluorescence) very graphically validates the kinesin pulling assay, could this be squeezed into the main text? In this instance, presumably all 3 kinesin molecules ultimately engaged?

Point-by-point response to reviewer comments

Reviewer #1:

The authors present single molecule experiments that investigate the force required to remove a tubulin subunit from the microtubule (MT) lattice. The study is very timely as multiple papers have emerged recently regarding MT lattice damage and repair by motors, and the implications of the study would be of interest of a wide biophysical audience. The authors present two innovative biophysical assays: site-specific labeling of alpha tubulin and the kinesin-DNA-tubulin pulling assay that could be adapted for other studies. Overall, the paper is well-written. However, I have some concerns that should be addressed before publication.

1. Title: The title is very broad. As pointed out throughout the paper, the authors investigate the force to remove tubulin from the MT lattice... but under very specific conditions. As noted in comments below, changing those conditions could have implications on the force required to remove the tubulin subunit. I suggest narrowing the scope of the title to fit the findings in the paper.

We have now changed the title to “The force required to remove tubulin from the microtubule lattice by pulling on its α -tubulin C-terminal tail” to specify the pulling point, as suggested.

2. Abstract: “...AAA-unfoldases, suggesting that severing proteins such as spastin and katanin use an unfoldase mechanism.” Maybe, but that statement is not conclusive from this study. I suggest scaling back the language.

The statement is removed from the abstract.

3. Abstract: “Our results reveal the response of tubulin to mechanical force and advance...” Very broad. Again, these experiments are under very specific conditions, and a generalized understanding of the force to displace a tubulin subunit from the MT lattice has remaining questions. Narrow the language here.

We have modified the abstract to qualify our conclusions, as suggested.

4. Results, Page 3, line 24: What do the authors mean by a labeling density of 43%. Is this saying 43% of all alpha-CTTs are labeled on a MT? Please clarify.

Yes, 43% of all alpha-CTTs are labeled. We clarified in the text.

5. Results, Page 4, line 17: The authors use the site-specific labeling (SSL) method to conjugate alpha tubulin CTTs with a single stranded oligo-DNA strand. Controls were performed to show that alpha CTTs labeled with a fluorophore via the SSL method do

not have large effects on MT dynamics. But why not do the same control with the oligo-DNA labeled tubulin? From the methods section, it sounds like the oligo-DNA tubulin, biotinylated tubulin, and the longer DNA linker were incubated together before polymerization (Methods, Page 20, starting at line 6). I would imagine that a large, floppy DNA linker could potentially affect MT dynamics. And at what labeling density of the small and large DNA causes differences in MT dynamics? A comparison between the oligo-DNA MT dynamics and full DNA tether-labeled MT dynamics could also give a clue regarding how those labeled tubulin get incorporated into the lattice. Perhaps local strain due to the large attachment and its Brownian motion “primes” that particular subunit to become displaced in the force pulling experiments.

We cannot do the suggested experiment with oligo-DNA labeled tubulin because we are limited by the labeling efficiency: the majority of the tubulin is unmodified as we stated in the text (the oligo tubulin labeling density is ~3%). We expect that the microtubules’ dynamic behavior will be governed by the large unlabeled tubulin fraction.

We included our finding that the SSL method doesn’t affect dynamics as a general statement about the technique rather than as being directly related to our results, but we think that it is useful information and want to keep it in the manuscript. While we agree that understanding the difference in the incorporation kinetics of the DNA-labeled tubulin may be of interest, we argue that the effect of DNA on microtubule dynamics is not directly relevant for our experiments because we do not expect Brownian motion of the DNA to produce additional forces on the tubulin beyond those exerted by the solvent molecules. We have now addressed the possible impact of DNA labeling in the Results.

6. Results, Figure 2 and Page 5, lines 10-11: I’m convinced of the integrity of all of the connections in the pulling assay except for the BSA/coverglass interaction. What forces can the BSA/coverglass interaction undergo before detaching from the surface?

We don’t see any evidence for the microtubule detaching from the surface as they were attached by a large number of biotinylated-BSAs. With a density of tens of attachments per micron, the trapping forces would be distributed over many BSA molecules and unlikely to induce detachment.

Also, how does the relative amount of biotinylated tubulin change the stiffness of the MT? I could imagine that more biotinylated tubulin within the MT lattice would cause more interactions with the neutravidin coated surface (assuming the biotinylated BSA stays put as assumed in the assay) and thus cause a tighter/stiffer interaction of the MT with the coverglass surface. This relative stiffness could also influence interactions within the MT lattice and maybe emphasize any “defects” or “alterations” to the normal lattice pattern that could be influenced by the DNA-labeled tubulin.

We do not expect the biotinylated tubulin to have a significant impact on the stiffness directly, as previous studies showed that small molecule labeling like fluorophores do not affect the microtubule stiffness (e.g., Gittes et al., J. Cell Biol. 1993). We do not expect the surface attachment to have an impact on our measurements, and this is borne out in that the tweezer and motor pulling assays used different attachment strategies (strong neutravidin-biotin vs. weaker anti-DIG antibody), but the results are not drastically affected. We have now discussed surface attachment as a possible caveat in the Discussion.

7. Results, Page 6, starting at line 4: The authors state that they only do the tweezers experiment by pulling perpendicular to the MT axis. “We chose this pulling direction mainly to avoid the attachment of more than one DNA linker to the bead during the pulling experiment.” While pulling perpendicular is of course interesting, it is possible that pulling direction along the axis of the MT could result in differing rupture force (including differences in plus vs minus end pulling, effects from pulling against the neighboring lattice subunits).

Further, perpendicular pulling direction may be relevant for potential mechanisms of severing enzymes, but not necessarily for motor induced damage while walking.

We agree with the reviewer that the force geometry can potentially result in different tubulin extraction forces. Also, different physiological processes are likely to be associated with different directions (e.g., severing perpendicular and motors axial). We now added an extensive discussion about the differences in the pulling directions and anchor points, addressing this and several related issues brought up by both reviewers. However, as we stated in the Discussion, while the geometries of the tweezer and motor assays differ, the force range is not drastically different, implying that the pulling angle may not have a large impact on the tubulin extraction force.

In order to pull in different directions could the following assay alterations be implemented to accomplish such pulling?

1. Use fluorescent polarity marked MTs (bright GMPCPP seed, dimmer elongation, taxol stabilized) and dilute the oligo-DNA tubulin concentration such that only one or two tethers can form per MT? That way you can pull either toward the plus or minus end with no interference from other tethers. Yes, that makes an already low throughput experiment more low throughput, but there are multiple MTs per coverglass that could be probed.

This is a good suggestion to reduce the number of oligo-DNA tubulin on the lattice. We tried this in early experiments but found that the number of successful attachments to the bead was very low so that we were unable to collect sufficient data due to the low throughput.

2. Or use polarity marked MTs but change the assay order. Incubate the anti-dig beads with the long DNA-dig strand first, then add to the assay with oligo-DNA MTs. You then actively form tethers by trapping the bead, bringing it to the surface, and grazing the surface of the MT until a connection is made. Then pull in the respective direction. This is another good suggestion which we also tried early on. However, we found that it is highly inefficient to form the tether by trapping the bead and actively approaching to the microtubules, and the successful attachment of the DNA was hard to judge because of the long linker length.

8. Results, Page 6, lines 20-21: It is not clear whether the authors know that a single tubulin subunit is being removed (just alpha tubulin, the whole tubulin dimer, more local damage?). The authors should comment on these possibilities or clarify their language. The reviewer is quite correct: we do not have direct evidence to distinguish whether one or multiple tubulins comes out. We now addressed this point in the Discussion.

Perhaps a MT-lattice repair assay with soluble labeled tubulin could help narrow down the level of damage induced.

This is a good suggestion, but significantly adds to the complexity of an already complicated experiment. Also, our tweezer setup does not have fluorescence.

9. Results, Page 7, lines 3-4: The authors state that a “complication of our optical trap assay is the uncertainty of the actual pulling direction with respect to tubulin...” I thought above the authors were pulling perpendicular to the MT axis? Please clarify.

We apologize for not making the geometry clearer. In the optical tweezer assay, we pulled perpendicular to the microtubule axis. However, the force is parallel to the plane of the surface and so the pulling direction with respect to the tubulin will be different depending on the azimuthal location of the tubulin. Thus, tubulins on top will be pulled tangentially by the trap, but those on the near side will be pulled perpendicularly. Those on the far side will be pulled in a complicated way. We now clarify the pulling direction in the tweezer assay in detail in the Results and modified Fig. 2 to add the x-, y-, and z-axes. The variability in direction relative to the attached tubulin dimer may lead to variability in forces: we have added this as a caveat in the Discussion.

10. Figure 4A: It appears that there's a smaller step size distribution at around 35 nm, but these are excluded from unfolding force analysis (page 8, line 23). Why?

Because the number of events was small (5 data points; <10% of the total events), we did not focus on these events: perhaps they are due to different geometries, or to an alternative unfolding pathway.

11. Figure 5C caption: “Time-lapse images of the last stretching round before rupture...”

What does “last stretching round” mean? Do the motors walk to the end of the DNA extension, recoil back after stall, and begin walking again several times (similar to a single molecule kinesin optical trapping assay)?

Correct. We have now changed the “last stretching round” to “last stretching event” for clarification.

Is it possible that the repeated pulling on the tubulin subunit when the kinesins reach stall help to “pry” the subunit out of place over time?

This is an interesting point. We did not observe any difference in the rupture time where the DNA was stretched only once or multiple times (Fig. S6D). Thus, we do not think there is a memory effect in this assay, perhaps because of the rapid refolding in the absence of force.

12. Figure 5: The authors should do a comparison of the kinesin pulling assay with taxol stabilized MTs since the trapping assay uses that form of stabilization.

We thank the reviewer for this suggestion and have now included the results of the motor pulling assay using taxol-stabilized microtubules (Fig. 5E, Fig. S6B,C). The lifetime is about twice that of unstabilized MTs, implying that taxol stabilizes against pulling forces. We have also compared the motor pulling assay and optical tweezer assays results in the context of taxol microtubules experiments (see Discussion and Appendix in Supplementary Information).

Reviewer #2:

This is important work. The authors use a published method to switch the C-terminal tyrosine of alpha tubulin for a reactive azo-tyrosine, validate its use for site-specific attachment of a DNA linker and microbead to the C-terminus of alpha tubulin in the lattice, and exert calibrated forces on it using either an optical trap or a small team of kinesins. They measure the force required to rupture the connection between the bead and the microtubule, and they show that the force required is (marginally) less than that required to break the chain of linkages when it is connected directly to the substrate, arguing therefore that the observed rupture events indicate tubulins being pulled from the GDP-taxol lattice. They also detect a force dependent unfolding event, likely of the alpha tubulin C-terminal H11-H12 hairpin.

To my knowledge this is the first direct measurement of the force required to extract tubulin from the lattice. There are a number of caveats to each experiment, which the authors carefully consider. The worst case scenario is that some or even all of the ruptures detected actually represent breakage of the chain of connectivity, rather than

extraction of the tubulin. But even if this were so, and the extraction force were thereby under-estimated, by setting limits these measurements remain extremely valuable to the field. The experiments are a remarkable technical achievement and the authors have done everything feasible to strengthen the case that the values obtained are correct.

The authors in their discussion acknowledge that much remains to be done (test different types and states of lattice, find out if one molecule or several are being dragged out of the lattice), but this paper contributes much-needed quantification in a currently murky area. I think it will be invaluable.

I have only minor comments.

1. Even in the abstract, the authors try to generalise their result to infer that tubulin-extracting AAA ATPases like katanin could plausibly work by unfolding tubulin and transporting the unfolded chain against a load. This point is worth mentioning but I'm not convinced it belongs in the abstract. At best the current data tell us one way in which katanin might in principle work. But if anything rather than supporting an 'unfold and pump the unfolded chain' model for tubulin extraction, the data say that tubulin extraction involves a preliminary partial (~10% of the primary sequence) unfolding event, after which the tubulin molecule jumps out of the lattice en bloc.

We removed the statement related to the unfoldases model of severing enzymes from the abstract, as suggested.

2. The inferred zero force stay-resident lifetime of 40s for GDP-tubulin in the undamaged lattice is hard to believe. If this rate of spontaneous loss of tubulin from the lattice occurs, wouldn't capped GDP microtubules in the absence of free tubulin self destruct?

This is a very good point. A 40-s lifetime would lead to a very rapid disintegration of GDP microtubules. We mentioned in the original text that this discrepancy could be avoided if tubulin extraction is a two-step process that occurs via a reversible, partial unfolding step (i.e., what we observe in our optical tweezer experiments). We have now elaborated on this model in the Appendix. The model can potentially account for the long lifetime of lattice-bound tubulin and suggests that the force-dependent rupture time extrapolated based on a single-step Dudko model (Fig. S7) will greatly underestimate the rupture time at low force.

Bending the GDP-tubulin lattice in microfluidic flow is reported to create and expand defects and taxol fully protects. There are modelling-simulation studies that assign definite stabilities (energies) to the lateral and longitudinal bonds stabilising heterodimers in the lattice. Are the energies inferred here consistent?

We calculated the tension caused by the filament bending: ~ 100 pN for typical small bends. While this is large compared to our rupture forces, the geometry and attachment point are quite different. We have commented on this in the Discussion.

3. The trapping work is done with GDP taxol microtubules, which as the authors discuss are more stable to mechanical damage than GDP-microtubules. Perhaps make the point therefore that the forces measured in the trapping experiment will tend to over-estimate the force required to extract tubulin from a GDP-only lattice?

We now included the motor pulling assay on taxol microtubules and found that taxol increases the rupture time by ~ 2 fold (Fig. 5E, S6B,C), which is consistent with the stabilization effect of taxol. We compared it the optical tweezer measurements in the context of the two-step model as well.

This is a strength of the approach. A little more detail on why measurements with capped GDP microtubules were intractable would be interesting, since this too may support metastability of strained GDP tubulin molecules in the lattice, which is the point of interest here.

We stated in the original text that we were unable to perform the optical tweezer assay on the non-stabilized GDP-tubulin directly because of their low stability and short-lifetime due to spontaneous breakage and depolymerization. However, in response to reviewer #1, we added the complementary experiment, namely motor pulling on taxol-stabilized MTs. This goes some way towards addressing this limitation.

4. Fig S5B&C (stepwise increase in pulling force detected by DNA dye fluorescence) very graphically validates the kinesin pulling assay, could this be squeezed into the main text? In this instance, presumably all 3 kinesin molecules ultimately engaged?

We thank the reviewer's suggestion and now included additional examples of stepwise increase of force in Fig. 5F, showing examples of both two and three steps. We expect all 3 kinesins ultimately engaged at least in the three-step examples.

REVIEWERS' COMMENTS

Reviewer #1 (Remarks to the Author):

The authors have sufficiently addressed my concerns.

Reviewer #2 (Remarks to the Author):

I have carefully read the author's responses to my comments and confirm they have addressed all my points satisfactorily.